# Factors impacting a patient's selection of an otolaryngologist

Katelin R. Keenehan [1*], Amanda G. Baanante[1], Arunima Vijay[2],
Gaayathri Varavenkataraman[3], Michele M. Carr[3]

**1** Jacobs School of Medicine and Biomedical Sciences, University at Buffalo, Buffalo, New York, United States of America, **2** Department of Otolaryngology, University of Florida College of Medicine, Gainesville, Florida, United States of America, **3** Department of Otolaryngology, Jacobs School of Medicine and Biomedical Sciences, University at Buffalo, Buffalo, New York, United States of America

* krkeeneh@buffalo.edu

## Abstract

### Objective

How a patient chooses a physician has not been evaluated in otolaryngology. Our purpose was to determine which factors are important for people in the selection of an otolaryngologist.

### Methods

This study is a cross-sectional survey study using Amazon Mturk. Demographic data was collected. Participants rated 24 factors related to choosing an otolaryngologist on a 10-point scale. Multiple choice questions queried preferences on a physician's social media, clothing, gender, and personality.

### Results

Out of 287 participants, 105 (36.6%) were female, 209 (72.8%) were between the ages of 25–40, and 252 (87.7%) identified as white. The majority of participants had primary residence in the southern US (N = 115, 38.3%) and were native-English speakers (N = 286, 99.7%). 254 (88.5%) had completed a Bachelor's degree, 232 (80.8%) were employed in healthcare, 237 (82.6%) had prior experience being treated by an otolaryngologist, and 275 (95.8%) rated their health status as good or better. The most important factors for respondents in choosing an otolaryngologist were "surgeon's explanation of their diagnosis and treatment", "the quality of their surgeon's office", "favorable office location", "the surgeon's ability to grasp their problem", "the surgeon's professionalism", and "the surgeon's physical examination". Regarding attire and personality, formal or business casual with a white coat was preferred (N = 124, 43.2%). Participants preferred surgeons with a combination of a professional and lighthearted personality (N = 145, 50.5%).

**Data availability statement:** All relevant data are within the manuscript and its Supporting Information files.

**Funding:** The author(s) received no specific funding for this work.

**Competing interests:** The authors have declared that no competing interests exist.

## Conclusion

The way in which an otolaryngologist engages with and examines a patient is most important to the lay public, although reputation carries some importance in choosing a doctor.

## Introduction

Patient-centered care is a growing focus in healthcare that has led to greater emphasis on data about patient experiences and opinions [1]. Factors that impact the selection of a physician have been explored in primary care, cardiology, and general, oncologic, and orthopedic surgery [2,3]. Patient selection of an otolaryngologist however, remains understudied.

Previous national survey work [4] found that patients considered a variety of factors when choosing a physician, including the physician's reputation, recommendation by family and friends, and insurance-plan participation. More recent reviews [5] have emphasized that patient choices are influenced by a interplay of patient characteristics (such as socioeconomic status, prior experience, health literacy) and provider and institutional characteristics (such as accessibility, structural features, and provider interpersonal skills). Additionally, in the digital era, the nature of physician selection is evolving. For example, the recent study by Bai *et al.* [6] demonstrated that patients' perceptions of physician competence and warmth, as gleaned from online reviews and physician images, significantly influence decision-making.

Despite the broader literature, there is a gap in our understanding of how patients select otolaryngologists. Prior work in this specialty is confined to topics such as physician attire and white coat-preference [7]. A better understanding of these factors may help otolaryngologists align their practices with patient preferences and strengthen the patient-physician relationship.

This study aims to address this gap by investigating which factors, ranging from physician communication, professionalism, office location, online presence, to social media, are of greatest importance in a patient's selection of their otolaryngologist. By using a cross-sectional survey of U.S. adults, we provide novel insight into the decision-making process in otolaryngology and how it may differ from broader physician-choice literature.

## Materials and methods

### Participants

This study received approval from the University at Buffalo Institutional Review Board (STUDY00007074). Data was collected anonymously from participants on Amazon's Mechanical Turk (MTurk), an online crowdsourcing platform which has been found to produce reliable results when compared to traditional data collection methods [8]. Study inclusion criteria were that participants were at least 18 years old, residents of the United States, and able to read and complete the survey in English. Exclusion criteria were surveys that were incomplete or failed the attention-check question.

Participants were recruited between March 24, 2023, and May 26, 2023. Informed consent was obtained electronically through completion of the survey, which included an introductory statement explaining the purpose of the study and affirming that participation was voluntary and anonymous.

## Data collection

Eligible participants provided demographic information including age, gender, race, native language, education level, household income, primary residence within the US, and health insurance status and type. Additionally, they reported their overall health status, prior experience with or without an otolaryngologist, and employment in healthcare.

Participants rated their preferences for 24 factors and attributes of otolaryngologists and clinics that influence their choice of provider. They utilized a Likert scale from 0 to 10, with 0 being the least important and 10 being the most important, to evaluate the factors shown in Table 1.

Participants were also asked to select their preferences for social media posts (scientific work, surgeries/patient outcomes, popular science, memes, personal life, or other), clothing (professional, professional with white coat, scrubs, scrubs with white coat, or no preference), gender (male, female, nonbinary, or no preference), and personality (professional, a combination of professional/lighthearted, or largely lighthearted).

## Statistical analysis

Statistical analyses were performed with SPSS 30.0 (IBM Corp, 2024) and R 4.4.2 (R Core Team, 2024). Survey results were reported using descriptive statistics. Multivariable ordinal regression analysis was used to identify predictors of

**Table 1. Survey questions.**

| |
|---|
| Surgeon's educational background (institution where they trained) |
| Years of experience the surgeon has |
| Length of time until surgeon has appointment availability |
| Friends/family recommending the surgeon |
| Reputation of surgeon's hospital or group |
| Marketing (billboards, ads) |
| Surgeon states that they use new surgical technologies or techniques |
| Surgeon's or practice's website |
| Surgeon's Twitter posts |
| Surgeon's Facebook posts |
| Surgeon's Instagram posts |
| Favorable office location |
| Ease of parking |
| Surgeon's online reviews (Healthgrades, Google reviews, Yelp, etc.) |
| Professionalism of office staff |
| Quality of the surgeon's office/hospital facilities and buildings |
| Surgeon's attire/clothing |
| Surgeon's gender |
| Surgeon's professionalism |
| How much time the surgeon spends with you |
| Ability of the surgeon to grasp your problem |
| Surgeon's physical examination of you |
| Your ability to see the exam performed by the surgeon (screen with a view of scope/ear exam) |
| Surgeon's explanation of your diagnosis and treatment options |

preference scores. A significance level of p < .05 was initially applied; however, to account for multiple comparisons among the nine predictors included in the model (age, gender, race, education, income, insurance, overall health status, health-care worker status, and prior otolaryngologist visits visits), a Bonferroni correction was implemented. This adjustment set the threshold for statistical significance at p < .0056.

## Results

### Demographics

Out of 300 respondents, 13 were excluded after failing the attention check, and the remaining 287 were included in the analysis. Participants represented all regions of the United States, with 115 (38.4%) indicating that their primary residence was in the southern US. 209 participants (72.8%) were between the ages of 25 and 40 years old, and 182 (63.4%) were male. The majority of participants (N = 286, 99.7%) were native-English speakers, and 252 (87.8%) participants identified as White or Caucasian (Table 2). 254 (88.5%) respondents held at least a Bachelor's degree, 275 (95.8%) respondents rated their health status as good, very good, or excellent, and 237 (82.6%) had previously been treated by an otolaryngologist. The most commonly reported annual household income was between $30-60k (N = 144, 50.2%), and the most commonly reported insurance type was Medicare (N = 214, 74.6%) (Table 3).

### Otolaryngologist preferences

Participants most valued a surgeon's explanation of their diagnosis and treatment, the quality of their surgeon's office, favorable office location, the surgeon's ability to grasp their problem, the surgeon's professionalism, and the surgeon's

**Table 2. Participant demographics.**

| Demographic | N (%) |
|---|---|
| **Age** | |
| <25 years old | 17 (5.9) |
| 25-40 years old | 209 (72.8) |
| 41-65 years old | 56 (19.5) |
| 65 + years old | 5 (1.7) |
| **Gender** | |
| Female | 105 (36.6) |
| Male | 182 (63.4) |
| **Race** | |
| White or Caucasian | 252 (87.8) |
| Black or African American | 16 (5.6) |
| American Indian or Alaska Native | 14 (4.9) |
| Asian | 1 (0.3) |
| Other | 4 (1.4) |
| **Language** | |
| English | 286 (99.7) |
| Other | 1 (0.3) |
| **Location** | |
| Midwest | 76 (25.3) |
| Northeast | 43 (14.3) |
| South | 115 (38.3) |
| West | 53 (17.7) |

**Table 3. Health and socioeconomic factors.**

| Factor | N (%) |
|---|---|
| **Education** | |
| Some high school or less | 1 (0.3) |
| High school diploma or GED | 16 (5.6) |
| Some college, but not degree | 7 (2.4) |
| Associates or technical degree | 8 (2.8) |
| Bachelor's degree | 186 (64.8) |
| Graduate or professional degree | 68 (23.7) |
| Prefer not to answer | 1 (0.3) |
| **Are You a Healthcare worker?** | |
| Yes | 232 (80.8) |
| No | 55 (19.2) |
| **Overall Health Status** | |
| Excellent | 46 (16.0) |
| Very good | 110 (38.3) |
| Good | 119 (41.5) |
| Fair | 12 (4.2) |
| Poor | 0 (0.0) |
| **Previous Otolaryngology Consult** | |
| Yes | 237 (82.6) |
| No | 50 (17.4) |
| **Income** | |
| <$30,000 | 18 (6.3) |
| $30,000-$60,000 | 144 (50.2) |
| $61,000-$90,000 | 96 (33.4) |
| $90,000+ | 29 (10.1) |
| **Insurance** | |
| Private/commercial | 55 (19.2) |
| Medicare | 214 (74.6) |
| Medicaid | 8 (2.8) |
| Military/veteran | 9 (3.1) |
| Prefer not to answer | 1 (0.3) |

physical examination. Gender, marketing strategies, and social media presence were considered less important in the selection of an otolaryngologist (Fig 1).

Participants preferred social media posts showcasing surgeries/patient outcomes (N = 173, 60.3%) and scientific work (N = 156, 54.4%) rather than content related to popular science, memes, or surgeon's personal lives.

Almost half of the participants (N = 124, 43.2%) in this study preferred a surgeon who wore formal or business casual clothing with a white coat. Although 31 (10.8%) respondents expressed no gender preference for their surgeon, 157 (54.7%) preferred male surgeons, and 98 (34.1%) preferred female surgeons. Participants also preferred surgeons with a combination of a professional and lighthearted personality (N = 145, 50.5%) or a strictly professional and formal personality (N = 125, 43.6%), over those with predominantly lighthearted/humorous personalities (N = 17, 5.9%) (Table 4).

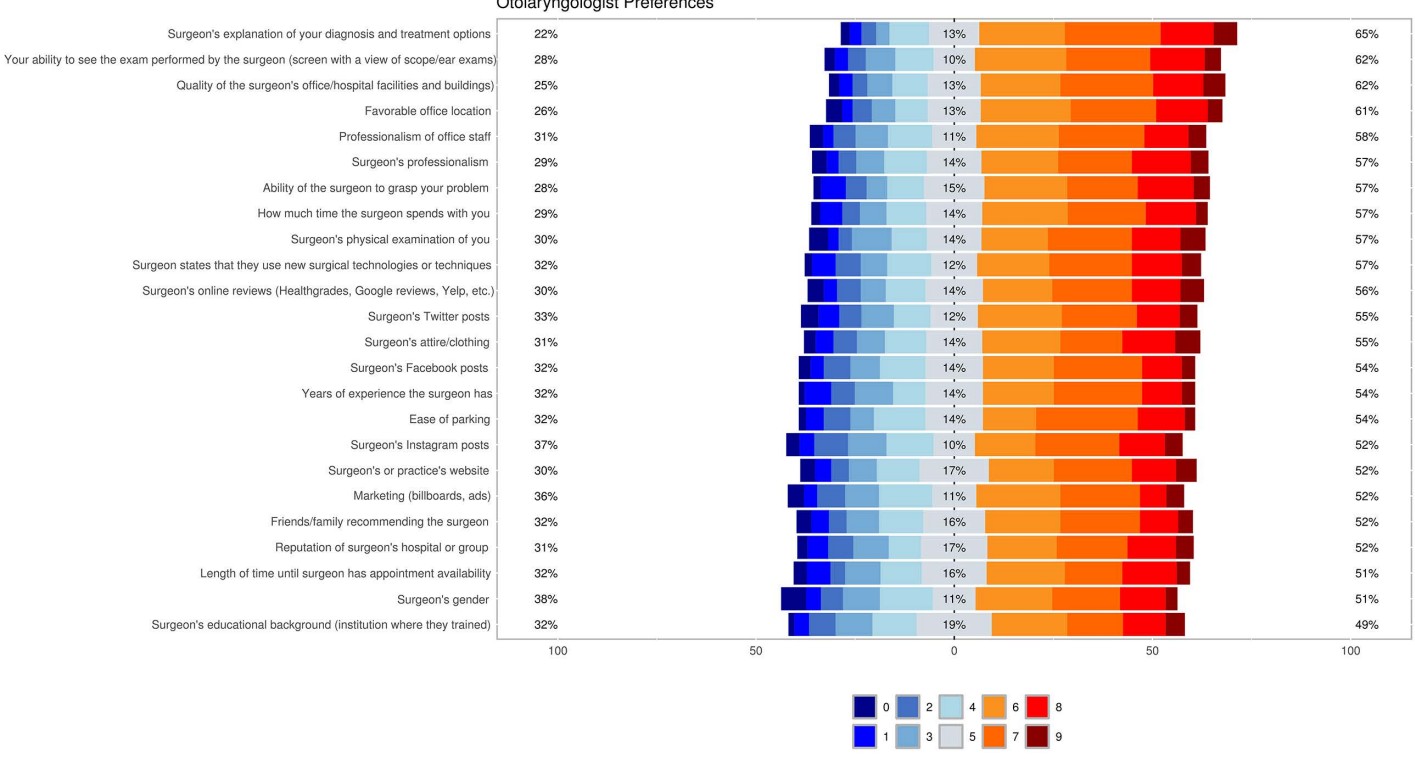

**Fig 1. Likert scale preferences.** Responses to 24 questions on a Likert scale of 0 (the least important factor) to 10 (the most important factor). Percentages on left represent participants who rated the factor as less important (0-4), percentages in center grey bars represent participants who rated the factor as neutral (5), and percentages on right represent participants who rated the factor as more important (6-10). Factors are listed from top to bottom in descending order of importance.

## Ordinal regression analysis of preference scores

All significant results of ordinal regression analysis are displayed in S1 Table. Ordinal regression revealed that older participants (65 + years old) were more likely to have higher preference scores for the ability of the surgeon to grasp the problem (β = 5.51, 95% CI = 2.91–8.12, p < .001), length of time until the surgeon has appointment availability (β = 4.30, 95% CI = 2.34–6.27, p < .001), and reputation of the surgeon's hospital or group (β = 3.65, 95% CI = 1.70–5.59, p < .001). Participants claiming higher incomes ($90,000+) were more likely to have higher preference scores for the years of experience the surgeon had (β = 2.39, 95% CI = 1.16–3.61, p < .001), the length of time until the surgeon had appointment availability (β = 1.93, 95% CI = 0.76–3.11, p = .001), marketing (β = 1.88, 95% CI = 0.65–3.11, p = .003), and the quality of the surgeon's office/hospital facilities and buildings (β = 1.88, 95% CI = 0.65–3.12, p = .003).

## Discussion

The findings from this cross-sectional survey suggest that communication and physical examination skills continue to be of high value to patients in their selection of an otolaryngologist. Effective communication between a patient and physician leads to increased trust and improved health outcomes, and performing a thorough physical exam limits oversights, misdiagnoses, or delayed treatment [9,10].

Participants in the study thought the most important factors to consider when choosing an otolaryngologist were quality of the office/hospital facility, communication, and physical exam. The quality of the surgeon's office and facilities

**Table 4. Participant preferences for otolaryngologist's social media activity, attire, gender, and personality traits.**

| Preference | N (%) |
|---|---|
| **Posts on scientific work** | |
| No | 131 (45.6) |
| Yes | 156 (54.4) |
| **Posts on surgeries/patient outcomes** | |
| No | 114 (39.7) |
| Yes | 173 (60.3) |
| **Posts on popular science** | |
| No | 166 (57.8) |
| Yes | 121 (42.2) |
| **Memes** | |
| No | 250 (87.1) |
| Yes | 37 (12.9) |
| **Posts on personal lives** | |
| No | 259 (90.2) |
| Yes | 28 (9.8) |
| **Attire** | |
| Formal or business casual | 85 (29.6) |
| Formal or business casual with white coat | 124 (43.2) |
| Scrubs | 26 (9.1) |
| Scrubs with white coat | 36 (12.5) |
| No preference | 16 (5.6) |
| **Gender** | |
| Male | 157 (54.7) |
| Female | 98 (34.1) |
| Non-binary/third gender | 1 (0.3) |
| No preference | 31 (10.8) |
| **Personality** | |
| Strictly professional/formal | 125 (43.6) |
| Combination of professional/lighthearted | 145 (50.5) |
| Largely lighthearted/humorous | 17 (5.9) |

was actually seen as a more important determinant than many of the patient-physician interaction factors explored. Engler *et al.* also found that the quality of the hospital or office facilities was an important factor for patients when choosing an orthopedic sports medicine surgeon, however, this factor did not outweigh surgeon professionalism and personality [2].

Although over 65% of physicians use social media [11] to support their professional practice, online engagement across all platforms were some of the lowest-rated factors influencing the selection of an otolaryngologist. Most of the highest-rated factors in our study were intrinsic to the patient visit, with online reviews rated as the most important factor beyond those specific to the visit. Based on a 2024 study looking at orthopedic surgeons' social media use, an increased number of patient reviews have only been found to be associated with active Twitter use [12], suggesting that Twitter may be the most valuable platform for otolaryngologists to use. These findings, however, are likely dependent on the ages of patients. Our results in this current MTurk study may be skewed by the fact that the majority of participants were under the age of 40.

Subjects in our study did not identify gender as being one of the more important factors when choosing an otolaryngologist; however, 54% preferred male surgeons when asked if they had a preference. A preference towards males also seems to exist amongst physicians referring their patients to surgeons. Dossa *et al.* identified that male physicians typically refer patients to male surgeons [13]. This disparity does not seem to be improving, despite the fact that more female physicians are becoming surgeons [13]. Due to these inequities, female surgeons have a lower patient volume when compared to their male counterparts [13]. This bias amongst patients and physicians seems to ignore the fact that female surgeons have lower rates of postoperative complications, as defined by Wallis *et al.* [14] When compared to patients of male surgeons, patients of female surgeons had lower rates of adverse outcomes, including death, both at 90 days and 1 year after surgery [14]. Further exploration is needed to address why bias exists against female surgeons and how to combat this issue in both American medicine and society.

Additionally, when considering a physician's gender, specialty-specific preferences may exist amongst patients and may be weighed more heavily in certain fields. For example, Tamalunas *et al.* looked at choosing a urologist and found that men referred to urologists significantly preferred to be treated by a male physician [15]. Similarly, more than 50% of women prefer to be seen by a female obstetrician/gynecologist [16]. Oftentimes dealing with the genitourinary system is sensitive for patients, and they may feel more comfortable being treated by a physician of the same gender, however, this may not be as important to patients looking for an otolaryngologist.

Patients also have preferences regarding their physician's attire. Li *et al.* showed that patients choosing a plastic surgeon preferred their physician to dress in formal wear with a white coat [17]. Petrilli *et al.* also found that patients prefer their physician to wear formal attire and a white coat when compared to scrubs and a white coat; however, respondents over 65 years old preferred scrubs for surgeons [18]. Almost half of our respondents were more likely to choose an otolaryngologist wearing formal or business casual attire with their white coat.

## Limitations

Though our study sought to characterize the factors that influence patient selection of their otolaryngologist, the study was limited to 287 participants, which may not reflect the preferences of all American otolaryngology patients. Many of our respondents stated that they were employed within healthcare, and this may also limit the external validity of the study. Additionally, the demographic composition of our sample, which was predominantly White, English-speaking, and highly educated, likely reflects the characteristics of Amazon MTurk users rather than those of the broader U.S. population. Prior studies have shown that MTurk participants tend to be younger, more educated, and less racially diverse than the general public [19,20]. As such, the findings of this study represent the preferences of MTurk users and may not be generalizable to the lay public. While this introduces a potential source of sampling bias, MTurk remains a validated and efficient platform for exploratory survey research [8]. Future studies should aim to recruit from multiple sources or employ stratified sampling to better capture diverse populations and perspectives.

## Conclusion

There are a variety of factors new otolaryngologists can consider to boost patient interest in their practice. Having a high-quality facility to practice in, communicating effectively by actively listening to the patient and explaining diagnoses or treatments clearly, and doing a thorough physical exam are factors patients feel are valuable and may contribute to developing a long-term patient-doctor relationship.

## Supporting information

**S1 Table. Significant predictors of patient preference scores identified by ordinal regression analysis at p < 0056.** Significant outcomes of ordinal regression analysis are displayed. The regression model included nine predictors: age, gender, race, education, income, insurance, health status, healthcare worker status, and previous ENT. To account

for multiple comparisons, a Bonferroni correction was applied, adjusting the significance threshold to $p < 0056$. Footnotes specify variable name and associated reference category. [a]Age; < 25 years old. [b]Gender; Male. [c]Race; White or Caucasian. [d]Education; High school diploma or GED. [e]Income; < $30,000. [f]Overall health status; Excellent health status. [g]Healthcare worker; Not employed in healthcare. Note: Insurance (Reference: Private/commercial) and Previous ENT (Reference: No previous ENT) were not significant predictors of any preference score.
(DOCX)

## Author contributions

**Conceptualization:** Katelin R. Keenehan, Amanda G. Baanante, Arunima Vijay, Gaayathri Varavenkataraman, Michele M. Carr.

**Data curation:** Katelin R. Keenehan, Amanda G. Baanante, Arunima Vijay, Gaayathri Varavenkataraman, Michele M. Carr.

**Formal analysis:** Katelin R. Keenehan, Amanda G. Baanante, Arunima Vijay, Gaayathri Varavenkataraman, Michele M. Carr.

**Investigation:** Katelin R. Keenehan, Amanda G. Baanante, Arunima Vijay, Gaayathri Varavenkataraman, Michele M. Carr.

**Methodology:** Katelin R. Keenehan, Amanda G. Baanante, Arunima Vijay, Gaayathri Varavenkataraman, Michele M. Carr.

**Resources:** Katelin R. Keenehan, Amanda G. Baanante, Arunima Vijay, Gaayathri Varavenkataraman, Michele M. Carr.

**Software:** Katelin R. Keenehan, Amanda G. Baanante, Arunima Vijay, Gaayathri Varavenkataraman, Michele M. Carr.

**Validation:** Katelin R. Keenehan, Amanda G. Baanante, Arunima Vijay, Gaayathri Varavenkataraman, Michele M. Carr.

**Visualization:** Katelin R. Keenehan, Amanda G. Baanante, Arunima Vijay, Gaayathri Varavenkataraman, Michele M. Carr.

**Writing – original draft:** Katelin R. Keenehan, Amanda G. Baanante, Arunima Vijay, Gaayathri Varavenkataraman, Michele M. Carr.

**Writing – review & editing:** Katelin R. Keenehan, Amanda G. Baanante, Arunima Vijay, Gaayathri Varavenkataraman, Michele M. Carr.

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
