## [Decision Letter · Decision Letter 0]

15 Sep 2025

Dear Dr. Keenehan,

ACADEMIC EDITOR:

Thank you for submitting your study about factors influencing patient selection of an Otolaryngologist. The topic is relevant and can help Otolaryngologists in improving their practice.

Introduction: Recommend improving the content and length. Refer  PMID: 12785569  and https://doi.org/10.1016/j.heliyon.2024.e28563

Materials and Methods: Participants were recruited from an online crowdsourcing platform. What were the inclusion and exclusion criteria for selection of participants?  How do authors explain the lack of diversity in the participants selected for the survey? 

Results: Although authors mention lack of diversity in the study as a limitation, the lack of diversity is critical to the external validity of the study. A cohort with a majority of white participants and majority healthcare employees is  not representative of the "lay public".   

Recommend addressing the bias directly and stating that the study only represents Amazon MTurk users and not the lay public. Alternatively, redesign the study to include a comparative cohort and use more demographically diverse sampling methods such as in-person surveys in the clinic or broader online platforms. 

We look forward to receiving your revised manuscript.

Kind regards,

Gauri Mankekar, MD,PhD,FACS

Academic Editor

PLOS ONE

Journal Requirements:

2. Please amend your authorship list in your manuscript file to include author Katelin Keenehan.

3. Please amend the manuscript submission data (via Edit Submission) to include author Katelin R. Keenehan.

Reviewers' comments:

Reviewer's Responses to Questions

**Comments to the Author**

1. Is the manuscript technically sound, and do the data support the conclusions?

Reviewer #4: Yes

Reviewer #5: Yes

Reviewer #6: Yes

Reviewer #7: Yes

2. Has the statistical analysis been performed appropriately and rigorously?

Reviewer #4: Yes

Reviewer #5: Yes

Reviewer #6: Yes

Reviewer #7: Yes

3. Have the authors made all data underlying the findings in their manuscript fully available?

Reviewer #4: Yes

Reviewer #5: Yes

Reviewer #6: Yes

Reviewer #7: Yes

4. Is the manuscript presented in an intelligible fashion and written in standard English?

Reviewer #4: Yes

Reviewer #5: Yes

Reviewer #6: Yes

Reviewer #7: Yes

Reviewer #4: The manuscript demonstrates a commendable level of methodological rigor and conceptual clarity throughout. This article offers useful insights for otolaryngologists and healthcare administrators seeking to align their practices with patient expectations. It lays a strong foundation for further qualitative research in diverse patient populations to enrich our understanding of these preferences across cultural and socioeconomic lines.

Reviewer #5: the article is clearly written and well-organized. Tables and figures support the text effectively and the language is professional and concise. This study contributes valuable knowledge, especially for otolaryngologists aiming to improve patient engagement.

Reviewer #6: There is bias because most of the patients considered in this study are from the healthcare industry and might be influenced by their knowledge / prior experiences of the physicians they wish to consult.

However their point is valid that when seeking consultation for areas that concern the urogenital tract , their choices tend to be restricted to person of their own gender

Reviewer #7: The manuscript is technically sound and the data support the stated conclusions. Statistical analysis has been applied appropriately. The data underlying the findings appear sufficiently presented, but a formal data availability statement would be advisable if not submitted already. The manuscript is well organized, intelligible. Minor clarifications in methods and figure presentation would enhance transparency. No concerns regarding dual publication or research ethics.

**Do you want your identity to be public for this peer review?** For information about this choice, including consent withdrawal, please see our Privacy Policy

Reviewer #4: **Yes: ** Abril Valentina Chiosso

Reviewer #5: **Yes: ** Francesca Santeusanio

Reviewer #6: **Yes: ** Christopher de Souza

Reviewer #7: No

---

## [Author Response · Author response to Decision Letter 1]

29 Oct 2025

Response to Reviewers

Manuscript ID: PONE-D-25-22757

Manuscript Title: Factors impacting a patient’s selection of an otolaryngologist

Response: Thank you for the opportunity to address the comments from the Editors and Reviewers at the PLOS ONE Editorial Office. The authors hope that the Editors and Reviewers will be satisfied with the changes we have made to the manuscript after considering their careful feedback.

Editor Comments:

Thank you for submitting your study about factors influencing patient selection of an Otolaryngologist. The topic is relevant and can help Otolaryngologists in improving their practice.

Response: We thank the Editor for their positive feedback and for recognizing the relevance of our study.

Introduction: Recommend improving the content and length. Refer PMID: 12785569 and https://doi.org/10.1016/j.heliyon.2024.e28563

Response: Thank you for this helpful suggestion. The Introduction has been expanded to provide additional context. We incorporated the supporting literature provided, as well as PMID: 22913549, in order to better situate our work within existing research and highlight the gap this study addresses.

Materials and Methods: Participants were recruited from an online crowdsourcing platform. What were the inclusion and exclusion criteria for selection of participants? How do authors explain the lack of diversity in the participants selected for the survey?

Response: Thank you for raising this point. We have now clearly stated the inclusion and exclusion criteria in the Materials and Methods section. Additionally, we have expanded the Discussion to address the limited diversity among participants, referencing prior studies that have characterized the MTurk population as generally younger, less racially diverse, and more educated compared to the general U.S. population.

Results: Although authors mention lack of diversity in the study as a limitation, the lack of diversity is critical to the external validity of the study. A cohort with a majority of white participants and majority healthcare employees is not representative of the "lay public". Recommend addressing the bias directly and stating that the study only represents Amazon MTurk users and not the lay public. Alternatively, redesign the study to include a comparative cohort and use more demographically diverse sampling methods such as in-person surveys in the clinic or broader online platforms.

Response: We thank the reviewer for this insightful comment. We have revised the Limitations section to explicitly acknowledge that the study sample reflects the characteristics of Amazon Mechanical Turk users rather than the lay public. We also included a statement that future studies could use more demographically diverse sampling methods or include comparative cohorts to improve external validity.

Reviewer #4 Comments:

The manuscript demonstrates a commendable level of methodological rigor and conceptual clarity throughout. This article offers useful insights for otolaryngologists and healthcare administrators seeking to align their practices with patient expectations. It lays a strong foundation for further qualitative research in diverse patient populations to enrich our understanding of these preferences across cultural and socioeconomic lines.

Response: Thank you for the positive feedback.

Reviewer #5 Comments:

The article is clearly written and well-organized. Tables and figures support the text effectively and the language is professional and concise. This study contributes valuable knowledge, especially for otolaryngologists aiming to improve patient engagement.

Response: Thank you for the positive feedback.

Reviewer #6 Comments:

There is bias because most of the patients considered in this study are from the healthcare industry and might be influenced by their knowledge / prior experiences of the physicians they wish to consult. However their point is valid that when seeking consultation for areas that concern the urogenital tract , their choices tend to be restricted to person of their own gender.

Response: Thank you for this feedback. We have acknowledged this as a limitation in the Discussion, consistent with the Editor’s request.

Reviewer #7 Comments:

The manuscript is technically sound and the data support the stated conclusions. Statistical analysis has been applied appropriately. The data underlying the findings appear sufficiently presented, but a formal data availability statement would be advisable if not submitted already. The manuscript is well organized, intelligible. Minor clarifications in methods and figure presentation would enhance transparency. No concerns regarding dual publication or research ethics.

Response: Thank you for this feedback. We have clarified the methods section to improve transparency.

---

## [Decision Letter · Decision Letter 1]

27 Nov 2025

Factors impacting a patient’s selection of an otolaryngologist

PONE-D-25-22757R1

Dear Dr. Keenehan

Thank you for revising the manuscript per reviewer's suggestions and comments. We’re pleased to inform you that your manuscript has been judged scientifically suitable for publication and will be formally accepted for publication once it meets all outstanding technical requirements.

Kind regards,

Gauri Mankekar, MD,PhD,FACS

Academic Editor

PLOS ONE

Reviewers' comments:

Reviewer's Responses to Questions

**Comments to the Author**

Reviewer #8: All comments have been addressed

2. Is the manuscript technically sound, and do the data support the conclusions?

Reviewer #8: Yes

3. Has the statistical analysis been performed appropriately and rigorously?

Reviewer #8: Yes

4. Have the authors made all data underlying the findings in their manuscript fully available?

Reviewer #8: Yes

5. Is the manuscript presented in an intelligible fashion and written in standard English?

Reviewer #8: Yes

Reviewer #8: The manuscript provides valuable insight into patient preferences and factors affecting the selection of an otolaryngologist. Consider including a line on implications for future research and otolaryngologist training programs. To improve clarity consider condensing data tables and avoid excessive demographic details.

**Do you want your identity to be public for this peer review?** For information about this choice, including consent withdrawal, please see our Privacy Policy

Reviewer #8: No

---

## [Editor Report · Acceptance letter]

PONE-D-25-22757R1

PLOS ONE

Dear Dr. Keenehan,

I'm pleased to inform you that your manuscript has been deemed suitable for publication in PLOS ONE. Congratulations! Your manuscript is now being handed over to our production team.

Kind regards,

on behalf of

Dr. Gauri Mankekar

Academic Editor

PLOS ONE